# Two-Photon Absorption and Multiphoton Excited Fluorescence of Acetamide-Chalcone Derivatives: The Role of Dimethylamine Group on the Nonlinear Optical and Photophysical Properties

**DOI:** 10.3390/molecules28041572

**Published:** 2023-02-06

**Authors:** André Gasparotto Pelosi, Eli Silveira-Alves, Leandro Henrique Zucolotto Cocca, João Victor Valverde, Guilherme Roberto Oliveira, Daniel Luiz da Silva, Leonardo De Boni, Pablo José Gonçalves, Cleber Renato Mendonca

**Affiliations:** 1São Carlos Institute of Physics, University of São Paulo, São Carlos 13566-590, SP, Brazil; 2Institute of Chemistry, Federal University of Goiás, Goiânia 74690-900, GO, Brazil; 3Department of Natural Sciences, Mathematics and Education, Federal University of São Carlos, Araras 13604-900, SP, Brazil; 4Institute of Physics, Federal University of Goiás, Goiânia 74690-900, GO, Brazil

**Keywords:** acetamide-chalcones, two-photon cross-section, SOS model, dimethylamine group, two and three-photon excited fluorescence emission

## Abstract

This work studied the effect of different electron-withdrawing and electron-donating groups on the linear and nonlinear optical properties of acetamide-chalcone derivatives. The results showed that the addition of the dimethylamine group led to a large fluorescence emission (71% of fluorescence quantum yield in DMSO solution) that can be triggered by two and three-photon excitations, which is essential for biological applications. Furthermore, dimethylamine also red-shifts the lower energy state by approximately 90 nm, increasing the two-photon absorption cross-section of the lower energy band by more than 100% compared with the other studied compounds. All compounds presented two-electronic states observed through one and two-photon absorption spectroscopy and confirmed by Quantum Chemistry Calculations (QCCs). QCC results were also used to model the experimental two-photon absorption cross-sectional spectrum by the Sum-Over-States (SOS) approach, revealing a dependence between the coupling of the ground state with the first excited state and the transition dipole moment between these states.

## 1. Introduction

In previous years, the incorporation of electron-withdrawing (EW) and electron-donating (ED) groups in organic molecules’ backbones has been driving a great variety of biological, photonic, and optoelectronic applications [1,2,3,4,5,6], owing to the ability of such groups to enhance the linear and nonlinear optical of the organic structure [7]. A well-known example of this molecular structure is EX-π-EX, where X can be a withdrawing or donating group and π is the organic backbone. Due to the charge transfer mechanisms, this type of molecular arrangement provides higher polarizability, improving the optical response. In this vein, it is fundamental to study new organic backbones with different EW and ED groups to find the optimal structures to improve emerging applications.

Chalcones, also known as α, β-unsaturated ketones, are present in an extensive range of optical applications, such as second harmonic generation [8,9,10,11,12,13], holographic recording technology [14], and photorefractive polymers [15], to name a few. In addition, owing to the intrinsic structural flexibility, functionality, and electron mobility of chalcones, modulation of third-order nonlinearities has been explored [1,2,4,16,17]. Moreover, when strong EW/ED groups are attached to the chalcone backbone, a high emissive state may be created, allowing applications involving DNA or metal ion detection [18]. In this regard, fluorescent-based chalcones are great candidates for fluorescence microscopy applications [19,20,21] because, besides the high biological activity [22] of chalcone derivatives, they present low interference with the target of interest [23]. Additionally, multiphoton fluorescence microscopy presents relevant advantages over single-photon ones, such as infrared excitation and high-resolution excitation, making compounds able to present fluorescence triggered by multiphoton absorption as a select class of compounds to be investigated.

Considering the importance of acetamide-chalcone derivatives, an experimental and theoretical study was performed on eight acetamide-chalcone derivatives with different withdrawing or donating groups. Measurements of linear absorption were performed for all compounds as well as a 2-photon absorption (2PA) cross-sectional spectrum. In the case of the compound with the dimethylamine group (the only fluorescent compound studied here), a high fluorescence quantum yield (71% in DMSO) was observed. A deeper spectroscopy investigation was performed for this compound, measuring the fluorescence emission, fluorescence quantum yield for different solvents, fluorescence lifetime, solvatochromism, and multiphoton excited fluorescence. Moreover, it was noted that there was a significant increase in the 2PA cross-sectional value (54 GM) at 800 nm for the first excited state as compared with the other studied compounds (18 GM). In addition, the Sum-Over-States (SOS) approach was applied to model the 2PA results and showed good agreement with the experimental results, revealing an important feature about the coupling of the first excited state with the ground state of the studied compounds. Finally, Quantum Chemistry Calculations (QCCs) were performed to obtain more details about the electronic structure of all studied compounds, such as permanent dipole moments, that were used as input parameters in the SOS approach.

## 2. Results and Discussion

The molecular structures of all compounds studied in this work are shown in Figure 1. All compounds have an acetamide group linked to ring A, the difference among them being the nature of the charge transfer and electron withdrawing/donating (EW and ED, respectively) features of the groups at ring B. Compounds **ChCH3**, **ChCH2CH3**, **ChOCH3**, **ChOCH2CH3**, and **ChN(CH3)2** present an ED group at ring B, while **ChBr** and **ChNO2** present an EW group. The basic structure, with no peripherical group at ring B, was also studied and named **ChH**. Details about the synthesis can be seen in the Materials and Methods section.

Figure 2 shows the molar absorption coefficient spectra (ε(λ)) (black lines) of acetamide-chalcone derivatives dissolved in DMSO. All electronic transition wavelengths (λ_0i_, where i represents the final electronic state) and molar absorption coefficient (ε_0i_) are shown in Table 1. Acetamide-chalcone derivatives were dissolved in DMSO (due to the good solubility for this solvent), and the UV-vis absorption spectra were acquired in different concentration ranges (2–50 × 10^−6^ M), as shown in Appendix A. ε spectra presented an intense absorption band (27,716–36,674 M^−1^ cm^−1^) ranging from 330 nm to 350 nm, depending on the compound. For compound **ChN(CH3)2**, the absorption band was centered at c.a. 420 nm. The high values of ε(λ) suggested that such a band could be ascribed to strongly allow π-π* electronic transitions. It should be highlighted that except for **ChN(CH3)2, ChOCH3** and **ChOCH2CH3** presented the most red-shift effects, indicating that these groups may be affecting their molecular planarity [24] and hence increasing the effective conjugation length, leading to a decrease in the energy of the electronic transition. In addition, the red-shift observed for these compounds could be associated with the non-bonding electrons of methoxy and ethoxy groups [25,26].

Although the linear absorption spectrum of the studied compounds exhibited only a single band (except for **ChN(CH3)2**), it was possible to note a shoulder at the blue side of the absorption band, indicating the existence of a second electronic excited state. This state was confirmed by using Gaussian decomposition on the ε spectra (see Appendix A). In addition, by performing TD-DFT calculations, we were able to identify the nature of the electronic states and simulate the absorption spectra, which agree considerably with the experimental ones. The results revealed a structured (asymmetric) absorption band (see Appendix A). The first transition was more intense and was found at c.a. 334 nm, and the second one had a lower intensity and was located at c.a. 285 nm. Furthermore, the average variances of the transition energy of the first and second bands of the theoretical results with the experimental ones were 0.014 and 0.027 eV, respectively. In this direction, from the Gaussian decomposition (Appendix A), the spectral position of both electronic transitions was determined (see Table 1), and transition dipole moments from the ground to the first and second excited states (μ_01_ and μ_02_, respectively), were calculated (see Table 2) by employing Equation (S1).

From Table 1, it is possible to note that the addition of electron-withdrawing/donating groups at the para-position of the B ring nearly affects the magnitude of ε related to the second excited state (ε_02_), ranging from 13,311 M^−1^cm^−1^ to 14,512 M^−1^cm^−1^. The first excited state (ε_01_) ranged from 21,665 M^−1^cm^−1^ to 35,169 M^−1^cm^−1^. Therefore, values of $\varepsilon_{01}$, $\varepsilon_{02}$ followed the same behavior in their value for compounds with overlapped electronic states, as well as for **ChN(CH3)2**, corroborating that the group **-N(CH_3_)_2_** is shifting the lower energy band. Therefore, the origin of the lower energy band may be related to a charge transfer process from the donor-acceptor group at the para-position of ring B to the keto-phenyl part through the π-bridge [27].

Two-photon absorption cross-section (δ^2PA^) spectra (see Figure 2, circles) were determined through the tunable femtosecond Z-Scan technique. The nonlinear absorption spectra were collected from 520 nm up to 750 nm with a 10 nm step. Two main two-photon peaks were observed along the nonlinear absorption spectra, which supported the interpretation that at least two excited states are allowed along such a spectral region. It is worth noting that the spectral position of 1PA, determined through Gaussian decomposition, overlapped with the measured peaks of the 2PA spectra. In addition, the second excited state had a higher 2PA cross-section magnitude than the first excited state for all compounds, with the exception of **ChN(CH3)2**, which presented the higher value of difference state dipole moment (see Table 2) and hence a higher value of nonlinear polarizability.

Results of 2PA cross-section maxima values are presented in Table 1 for two electronic states that are allowed by two-photon absorption processes. The 2PA lower energy band was centered at around 700 nm, except for **ChN(CH3)2**, which was at c.a. 840 nm, while the higher energy one was around 630 nm, again, except for **ChNO2** which was at c.a. 555 nm. For the second excited state, values of the 2PA cross-section ($\delta_{S2}^{2PA}$) ranged from 28 GM to 53 GM. However, among all the studied compounds, only **ChOCH2CH3** presented a $\delta_{S2}^{2PA}$ higher than 40 GM. The considerable dispersion between the 2PA values between the compounds may be related to the electron-withdrawing/donating groups bonded in the main conjugated molecular structure.

The results obtained in this work for the 2PA cross-section can be compared with those reported by Abegão et al. [4], also obtained using fs-laser pulses for unsubstituted and mono-substituted chalcone derivatives. For mono-substituted chalcone derivatives, the bonded group was **-OCH_3_** at the para-position, similar to the compound **ChOCH3** in this work. They obtained a 2PA cross-section of 14 GM c.a. 630 nm, which is two times lower than the **ChOCH3** compound, showing that the presence of an acetamide group at ring A led to an increase in the 2PA cross-section. For the reported unsubstituted chalcone [4], the value was even smaller (9 GM), which, when compared to the para-unsubstituted compound (**ChH**), was three times lower. Santos et al. [28] also reported values of 24 GM and 17 GM for the 2PA cross-section of brome and chlore substituted dibenzylideneacetone, which were very similar to chalcone derivatives in terms of structure, while for **ChBr,** a value of 33 GM was measured, reinforcing the importance of the acetamide group at ring A. Custódio et al. [6] also reported chalcone derivatives with a maximum 2PA cross-section of 17 GM c.a. 630 nm, which is at least two times lower than the value of the compounds studied in this work.

Besides 1PA and 2PA studies, QCCs were employed to visualize the electronic charge density of the Highest Occupied Molecular Orbital (HOMO) and the Lowest Unoccupied Molecular Orbital (LUMO) of acetamide-chalcone derivatives in order to investigate the nature of transitions. Figure 3 displays the frontier molecular orbitals (FMO) for all compounds. The lowest energy band for all molecules was described predominantly by the HOMO → LUMO excitation (ca. 83%) and the highest energy band by the HOMO-1 → LUMO excitation (ca. 83%) (See Appendix A), both having π→π* nature. This corroborates the initial analysis that the **-N(CH_3_)_2_** group led to a large red-shift of the lower energy band. The HOMOs of the compounds **ChH**, **ChCH3**, **ChCH2CH3**, and **ChBr** were very similar, showing a delocalization of the π-electrons over the entire scaffold. From the LUMOs, it was noted that the electrons had shifted to the carbonyl in the center, which was also observed for other chalcone derivatives [1]. Something similar happened for the compounds **ChOCH3** and **ChOCH2CH3**, except for the smaller electronic density distribution around ring A in the HOMO orbital. The HOMO → LUMO excitation of the **ChN(CH3)2** and **ChNO2** molecules involved a more pronounced electronic charge transfer. This feature was evidenced by the substantial electronic charge density around ring B in HOMO for **ChN(CH3)2**, revealing the strong ability to donate electrons of the **-N(CH_3_)_2_** group and, in LUMO, an electronic density distribution over the molecule structure. On the other hand, the compound **ChNO2** presented an electronic charge transfer from A to ring B due to the electron-withdrawing strength of the **-NO2** substituent. The compounds showed a decrease in the HOMO-LUMO energy gap concerning the **ChH** molecule. This justified the increase in the 2PA, as a smaller HOMO-LUMO gap makes the compound more polarizable and, consequently, improves the nonlinear responses [29]. Furthermore, due to the high polarizability and intramolecular charge transfer process, it was noted that their FMOs (HOMO/LUMO) had a large overlap, resulting in higher oscillator strengths [30] (see Appendix A).

It is worth describing that **ChN(CH3)2** is the only compound that shows fluorescence emission. These measurements were made in all compounds dissolved in DMSO at 10^−6^ M concentration. This fact may be understood because the dimethylamine group (**-N(CH_3_)_2_**) at the para-position of the B ring increases intramolecular charge transfer processes [18,23,27,31,32]. Such a group, owing to its strong electron-donor character, induces the unexpected formation of a huge dipole moment in the excited state, leading not only to a charge and molecular rearrangement of the solute but also to a structural rearrangement then creating an emissive intramolecular charge transfer (ICT) state.

Taking advantage of the fluorescence emission of compound **ChN(CH3)2**, fluorescence anisotropy (r), represented by the red circles in Figure 4, was measured for **ChN(CH3)2**. It is important to note that a constant value of r along the lower energy band indicates that in this spectral region, only one electronic state is being reached [33]. In this way, the strong electron-donor character of **-N(CH_3_)_2_** allows a large red-shift of the lower energy band, making it possible to observe both absorption bands separated when compared to the other compounds. In addition, to confirm the nature of the observed fluorescence signal of **ChN(CH3)2**, the inset of Figure 3 shows that the fluorescence spectrum position is independent of the excitation wavelength, according to ‘Kasha’s rule’ [34,35].

To obtain more insight into the features of the emissive ICT state of **ChN(CH3)2**, measurements of fluorescence quantum yields (*ϕ*) in different solvents are shown in Figure 5 as a function of solvent polarity (expressed in ET(30) [36]). An increase of the *ϕ* was observed with the solvent polarity ranging from 10% in toluene solution (non-polar solvent) to 81% in DMF solution (polar solvent), i.e., it presented a negative solvatokinetic effect. It is worth noting that in DMSO, the solvent used for the nonlinear optical measurements, the value of ϕ was 71%. For solvents such as ethanol and methanol, the fluorescence quantum yield exhibited a strong reduction in the fluorescence quantum yield (positive solvatokinetic) of 4% and 1%, respectively. The observed low values of *ϕ* for non-polar solvents (toluene, for instance) could be explained by the proximity effect of nπ* and ππ* states, which may destabilize the emitting state, hence quenching the fluorescence emission [31,37]. On the other hand, for the negative solvatokinetic effect, molecular conformational changes may be the main reason for increasing the values of *ϕ*. For alcohols solvents, such as ethanol and methanol, the ability to form hydrogen bonds between the solvent and the solute leads to fluorescence quenching owing to the deactivation of the lone pair of **-N(CH_3_)_2_** [38], elucidating the low values of *ϕ* in these solvents.

Additionally, measurements of the linear absorption and fluorescence emission in different solvents (Figure 6a,b, respectively) were made for **ChN(CH3)2**. The results showed a strong dependence of the spectral peak position of the fluorescence emission with the solvent polarity. Therefore, a considerable increase of the Stoke Shift was observed with the solvent polarity (from 77 nm in toluene solution up to 126 nm in methanol solution). A large bathochromic shift for polar solvents elucidated the strong ICT character of the first excited state. The permanent dipole moment difference (Δμ_01_) in DMSO was calculated via the Lippert-Mataga equation [39,40]:(1)$$\Delta \nu =\frac{\Delta \mu_{01}^{2}}{hca^{3}}\Delta f$$
in which Δν is the Stoke Shift for a certain solvent, h is the Planck constant, c is the light speed, “a” is the Onsager cavity radius, and Δf is the Onsager polarity function, given by Δf = 2((ε − 1)/((2ε + 1)) + (n^2^ − 1)/((2n^2^ + 1))), in which ε and n are the dielectric constant and refractive index of the solvent in question. The slope of Δν/Δf (Figure 6c) makes it possible to determine Δμ_01_ knowing the Onsager cubic radius a^3^. In this way, the only missing parameter is the Onsager cavity radius, which was determined through the Smoluchovski-Einstein equation [41]:(2)$$Vol=\frac{\tau KT}{\eta \left(\frac{0.4}{r}-1\right)}$$
where Vol is the spherical hydrodynamic volume occupied by the molecule in the surrounding medium, K is the Boltzmann constant, T is the room temperature, η is the solvent viscosity (1.996 cP for DMSO), r is the anisotropy coefficient, and τ is the fluorescence lifetime, which was determined through time-resolved fluorescence (TRF) measurements (Figure 6d—circles).

The results showed an experimental value of Onsager cubic radius ($a_{exp}^{3}$) of $a_{exp}^{3}=472A^{3}$, while QCC using PCM ($a_{G}^{3}$) resulted in $a_{G}^{3}=449A^{3}$, displaying a good agreement between experimental and theoretical values (a difference less than 5%). In this way, the permanent dipole moment difference was calculated (using $a_{exp}^{3}$), and a value of Δμ_01_ = 7.1 D was found, which is in agreement with the one obtained from the QCC (see Table 2). The large value of Δμ_01_ is owing to the strong electron-donating character of the **-N(CH_3_)_2_** group, leading to a high charge density distribution on the first excited state, as seen in its FMO (see Figure 2). Besides the importance of understanding the charge transfer mechanism, the value of Δμ_01_ was also used as an input parameter in the phenomenological SOS approach to better model the 2PA results.

To obtain more information about the electronic properties of the studied compounds, 2PA spectra were modeled by the Sum-Over-States (SOS) approach [42] (Figure 1, red lines), aiming to connect molecular parameters determined through linear spectroscopy with the nonlinear absorption spectrum (see Equation (3)). As confirmed by 1PA, 2PA, and QCCs, there are two electronic states that contributed to the absorptive nonlinearities. In this way, to model the experimental 2PA results, we have used:(3)$${\sigma \left(\omega \right)}_{2PA}=\frac{128\pi^{5}}{5{\left(hcn\right)}^{2}}L^{4}\left[\frac{\left(\Delta \mu_{01}^{2}\mu_{01}^{2}\right)G_{01}}{{\left(\omega_{01}-2\omega \right)}^{2}+G_{01}^{2}}+\frac{\left(\Delta \mu_{02}^{2}\mu_{02}^{2}\right)G_{02}}{{\left(\omega_{02}-2\omega \right)}^{2}+G_{02}^{2}}+\frac{\omega^{2}}{{\left(\omega_{01}-\omega \right)}^{2}+G_{01}^{2}}\frac{\left(\mu_{12}^{2}\mu_{01}^{2}\right)G_{02}}{{\left(\omega -2\omega \right)}^{2}+G_{02}^{2}}+\frac{\left(\Delta \mu_{02}\mu_{02}\mu_{01}\mu_{12}\right)G_{02}}{{\left(\omega_{02}-2\omega \right)}^{2}+G_{02}^{2}}\right]$$
where G_01_ and G_02_ are half-width at half-maximum of the first and second excited states, respectively, and ranged from 0.21 eV to 0.29 eV. μ_01_ and μ_02_ are the transition dipole moments from the ground state to the first and second excited states, respectively. μ_12_ is the transition dipole moment from the first excited state to the second one, Δμ_01_ and Δμ_02_ are the state dipole moment differences, ω_01_ and ω_02_ are the transition frequencies, and ω is the laser frequency.

To minimize the number of free parameters of the SOS approach and better fit the 2PA results, the following procedure was employed. Transition dipole moments (μ_01_ and μ_02_) were obtained through the Gaussian decomposition method, while Δμ_01_ and Δμ_02_ were determined with QCCs, with the exception of **ChN(CH3)2** because of its fluorescent nature, whose Δμ_01_ could be experimentally determined, as discussed earlier. In this way, the only free parameter in the SOS model was μ_12_. Results of μ_01_, μ_02_, Δμ_01_, Δμ_02_, and μ_12_ can be seen in Table 2. As we can note from Figure 2, the obtained values of Δμ_01_ and Δμ_02_, from QCCs, and μ_01_ and μ_02_, from Gaussian decomposition, well describe the experimental 2PA results (Figure 2—red lines). In addition, it should be mentioned that it is a rough task to determine μ_12_ experimentally, making the SOS approach an excellent tool for finding it.

The results of the QCCs revealed that the permanent dipole moment difference ($\Delta \mu_{01}$) for compounds **ChH**, **ChCH3**, and **ChCH2CH3** were 2.5 D, 4.6 D, and 4.8 D, respectively, while for compounds with oxygen atoms (**ChOCH3** and **ChOCH2CH3**) were 3.1 D, revealing that the oxygen atom may be decreasing the charge density distribution in the excited state. In addition, compounds **ChNO2** and **ChN(CH3)2** also displayed large Δμ_01_ because of the strong electron-donor character of this group, while **ChH** was the lower one due to the absence of EW/ED groups.

Another important feature elucidated by the SOS approach was the photophysical parameter μ_12_, which presented lower values for compounds that presented higher shifts between electronic states S1 and S2. For instance, compounds such as **ChOCH2CH3**, **ChNO2**, and **ChN(CH3)2** presented values ranging from 1.4 D to 3.7 D. This result was expected as the first excited state and the second one for these compounds was clearly shifted with respect to the other compounds, leading to a decrease in the excited states coupling. For other compounds for which both excited states overlapped, μ_12_ presented a low dispersion ranging from 4.3 D to 5.8 D.

As mentioned before, fluorescent chalcones have been extensively used in a wide range of biological applications. In this way, fluorescent chalcones presenting a fluorescence signal triggered by 2 and 3 photon excitations (2PE and 3PE, respectively) in the infrared spectral region are of interest for applications that require light excitation at the therapeutic window. Thus, fluorescence induced through 2PE (black circles—Figure 7) and 3PE (red circles—Figure 7) were analyzed for compound **ChN(CH3)2**, employing a femtosecond pulse laser with excitation wavelength (λ^ex^) at 900 nm and 1190 nm, respectively. It should be mentioned that the ability to fluoresce via 3PE has not been reported so far for similar compounds, as far as we know. Additionally, the low toxicity of compounds similar to **ChN(CH3)2** was already reported [43,44], which is essential for biological applications. Thus, the fluorescence was collected by an optical fiber for different excitation pulse energies, and plotted in a log-log scale, revealing an angular coefficient equal to 2 (for 2PE) and 3 (for 3PE), thus confirming that the fluorescence emission was triggered by 2PE and 3PE.

The active 2PA cross-section (product between the 2PA cross-sectional peak and the fluorescence quantum yields in DMSO solution) was calculated, which is also known as 2PA brightness, showing a value of 40 GM. To the best of our knowledge, such results have never been achieved for similar compounds. However, it is known that compounds with slightly lower 2PA brightness have been employed for biological imaging applications. The combination of all results involving compound **ChN(CH3)2**—green emission, Stokes Shift, high 2PA cross-section, efficient 2PE and 3PE at the therapeutic window, and high value of 2PA brightness—make it a candidate to be tested as a fluorescent bioprobe.

## 3. Materials and Methods

Acetamide-chalcones were synthesized through the Claisen-Schmidt reaction by adapting the methodology described in Ref. [45]. In a three-neck, round-bottom flask, we added 1 mmol of 4′-acetoamidoacetophenone with the corresponding 1 mmol benzaldehyde dissolved in a 5% NaOH/Ethanol solution at room temperature and stirred overnight. Reagents and solvents were used without any previous purification step, being purchased from Sigma-Aldrich *(*San Luis, CA, USA), Acros-Organics (Thermo Scientific, San Jose, CA, USA), and Neon (Suzano, Brazil). The progress of the reactions was followed through thin layer chromatography (TLC) using a hexane:ethyl acetate (85:15) solution as eluent (80:20). After the end of the reaction, HCl was added in an equimolar proportion to neutralize the NaOH, and the resultant solid was washed and filtered with ice water at 6 °C. The products were purified by recrystallization from a mixture of hexane:ethyl acetate (60:40) or methanol.

The final structures were built by condensation of acetamide-acetophenone and different benzaldehydes with electron acceptor and electron donor substituents. Table 3 summarizes all chalcones studied in the present work.

The structures of the obtained chalcones were confirmed by infrared (FTIR) and ^1^H and ^13^C NMR spectroscopies. FTIR spectra (Appendix A) were collected using KBr pellets of samples by transmission measurements from 4000 to 400 cm^−1^ on a Perkin-Elmer Spectrum 400 Spectrometer (Perkin Elmer Inc., Waltham, MA, USA). ^1^H and ^13^C NMR spectra (Appendix A) were recorded on a Bruker Avance III 500 MHz (11,75 T) spectrometer (Bruker Corporation, Billerica, MA, USA). The samples were solubilized in deuterated dimethylsulfoxide (DMSO-*d6—*Sigma Aldrich, San Luis, CA, USA*)* and deuterated chloroform (CDCl_3_—Cambridge Isotope Laboratories, Teksburry, MA, USA). HRESI-MS spectra were obtained using a Q Exactive hybrid Quadrupole-Orbitrap mass spectrometer (Thermo Scientific, San Jose, CA, USA). More details about FTIR, NMR, and HRESI-MS spectroscopies can be seen in Appendix A.

The nomenclature of the eight acetamide-chalcone derivatives studied in this work are:**ChH:** N-[4-[(2E)-1-Oxo-3-phenyl-2-propen-1-yl]phenyl]acetamide,**ChCH3:** N-[4-[(2E)-3-(4-Methylphenyl)-1-oxo-2-propen-1-yl]phenyl]acetamide,**ChCH2CH3:** N-[4-[(2E)-3-(4-Ethylphenyl)-1-oxo-2-propen-1-yl]phenyl]acetamide,**ChOCH3:** N-[4-[(2E)-3-(4-Methoxyphenyl)-1-oxo-2-propen-1-yl]phenyl]acetamide,**ChOCH2CH3:** -[4-[(2E)-3-(4-Ethoxyphenyl)-1-oxo-2-propen-1-yl]phenyl]acetamide,**ChN(CH3)2:** N-[4-[(2E)-3-[4-(Dimethylamino)phenyl]-1-oxo-2-propen-1-yl]phenyl]acetamide,**ChBr:** N-[4-[3-(4-Bromophenyl)-1-oxo-2-propen-1-yl]phenyl]acetamide,**ChNO2:** N-[4-[(2E)-3-(4-Nitrophenyl)-1-oxo-2-propen-1-yl]phenyl]acetamide.


**Linear optical measurements**


Linear absorption and fluorescence emission measurements were performed at concentrations of 10^−6^ M and placed in a quartz cuvette. For linear absorption and fluorescence emission measurements, a spectrometer UV-VIS Shimadzu 1800 (Shimadzu Corporation, Kyoto, Japan) and fluorimeter Hitachi F-7000 (Hitachi High-Technologies, Tokyo, Japan) were employed, respectively.


**Fluorescence Anisotropy measurements**


To determine the anisotropy coefficient, a fluorimeter Hitachi F-7000 (Hitachi High-Technologies, Tokyo, Japan) was employed. Fluorescence anisotropy is given by r = (I_vv_ − GI_vh_)/(I_vv_ + 2GI_vh_), and their values can vary in a range between −0.2 (if transition dipole moment is perpendicular to emission dipole moment) and 0.4 (if transition dipole moment is parallel to emission dipole moment). The indexes hh, hv, vh, and vv are related to the emission and excitation polarization channels (v = vertical and h = horizontal), which can be controlled through the use of two polarizers. In the fluorescence anisotropy definition, G = I_vh_/I_hh_ is associated with the sensibility of the emission and excitation channels of the fluorimeter, and I is the emission intensity for different excitation wavelengths (excitation spectrum). Figure 4 shows the absorption spectra (black line) and fluorescence anisotropy (red circles) for **ChN(CH3)2** dissolved in DMSO.


**Time-resolved fluorescence technique**


To collect the fluorescence lifetime signal, we employed a femtosecond laser pulse (CPA 2001, Clark MXR) at 385 nm (second harmonic of fundamental) as excitation. The fluorescence signal was collected perpendicularly to the excitation through an optical fiber connected to a photodetector. A digital oscilloscope was employed to visualize the signal. A mathematical convolution was applied to separate instruments artifacts from the real fluorescence signal, using the following procedure: *I*_measured_ = *I**_real_* ∗ *IRF* (with *I*_measured_ being the measured fluorescence signal, *I**_real_* the real fluorescence decay and *IRF* refers to the instrument response function). Therefore, a reference signal without any sample fluorescence was taken to collect the IRF signal.


**Tunable Femtosecond Z-Scan technique**


To measure the two-photon absorption cross-section spectrum, the tunable femtosecond open-aperture Z-Scan technique was employed. For this purpose, a femtosecond laser (CLARK, MXR^®^) centered at 775 nm with a pulse duration of 150 fs and a 1 kHz repetition rate pumps the Topas-Quantronix optical parametric amplifier (Light Conversion^®^) generating pulses from 460 nm to 2600 nm. This way, it was possible to determine the 2PA cross-section from 530 nm to 760 nm for all chalcone derivatives. For more details about the experimental setup, see Refs. [46,47,48].


**Quantum Chemistry Calculations**


Quantum chemistry calculations (QCC) were performed to corroborate the experimental analyses using a Gaussian 09 package [49]. For this purpose, analyses were performed at the theoretical level of the density functional theory (DFT) to obtain the optimized geometry of the molecules in the ground state using the hybrid B3LYP functional [50] combined with the 6-311G(d,p) basis set [51]. Vibrational frequency calculations were performed along with this calculation to confirm that minimum structures were reached [52]. Subsequently, the optimized structures were used to perform the time-dependent density functional theory (TD-DFT) calculations to determine the 15 lowest-energy singlet electronic transitions and the permanent dipole moments of the ground (**μ**_00_) and excited states (**μ**_kk_, k > 0), with which the difference between the dipole moments (**∆μ**_0k_= μ_kk_ − **μ**_00_, k > 0) was estimated [1]. For this purpose, the CAM-B3LYP functional [53] was combined with the 6-311++G(d,p) basis set. In the base set, 6-311G(d,p) was added to the diffuse function “++” to better describe electronic transitions [54]. Some of the side substituents can induce the molecules to have a charge transfer characteristic; therefore, it was chosen to use the long-range corrected CAM-B3LYP functional in the TD-DFT calculations [55]. In addition, all calculations were performed both in a vacuum and in a dimethylsulfoxide (DMSO) solvent medium using the polarizable continuous model (PCM) [56].

## 4. Conclusions

This work studied the role of different EW and ED groups on the linear and nonlinear optical properties of eight acetamide-chalcone derivatives. Two excited states were observed for this class of compounds, and both of them can be accessed by 1PA and 2PA processes. The existence of these states was corroborated through QCCs and the Gaussian decomposition method. The coupling between these states strongly depends on the EW or ED linked to the main molecular structure; such features were elucidated and quantified by the SOS approach by determining transition dipole moments between the first and second excited states. The incorporation of dimethylamine groups revealed an excellent path to promote a significant enhancement of linear and nonlinear optical properties in chalcone derivatives. An example is an increase of 100% in the 2PA cross-section at the lower energy band and the creation of a highly emissive excited state (71% of fluorescence quantum yield in DMF and DMSO). Finally, fluorescence triggered by 2PE ($\lambda_{ex}=900\;nm$) and 3PE ($\lambda_{ex}=1190\;nm$) were conducted, showing great potential for dimethylamine-chalcone derivatives as multiphoton fluorescent probes.

## Figures and Tables

**Figure 1 molecules-28-01572-f001:**
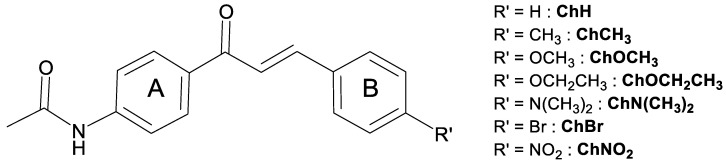
Molecular structure of the compounds studied. R’ represents the position of different substituents.

**Figure 2 molecules-28-01572-f002:**
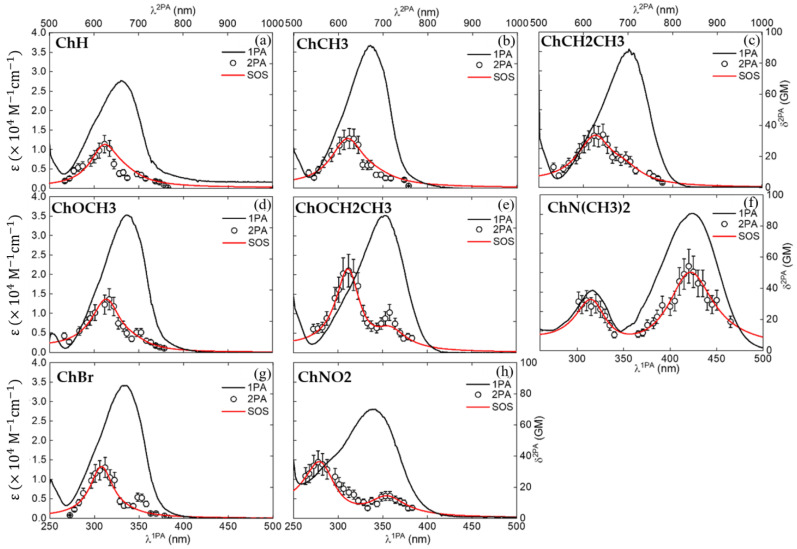
One-photon absorption spectrum (continuous black lines) and 2PA spectrum measured through the tunable Z-Scan technique (circles) of (**a**) **ChH**, (**b**) **ChCH3**, (**c**) **ChCH2CH3**, (**d**) **ChOCH3**, (**e**) **ChOCH2CH3**, (**f**) **ChN(CH3)2**, (**g**) **ChBr,** and (**h**) **ChNO2**. The experimental error associated with 2PA measurements is 20%. The red lines represent the SOS model used to model the experimental 2PA results.

**Figure 3 molecules-28-01572-f003:**
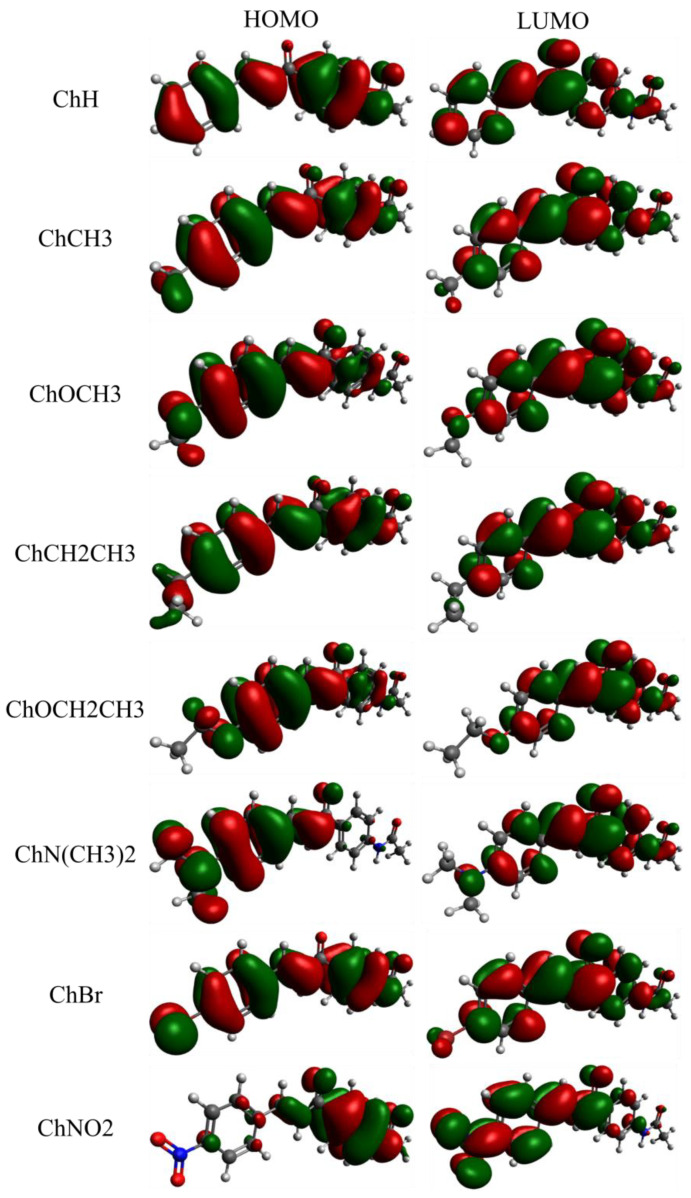
Frontier Molecular Orbitals of all studied compounds obtained through the PCM-CAM-B3LY/6-311++G(d,p) approach for DMSO solvent.

**Figure 4 molecules-28-01572-f004:**
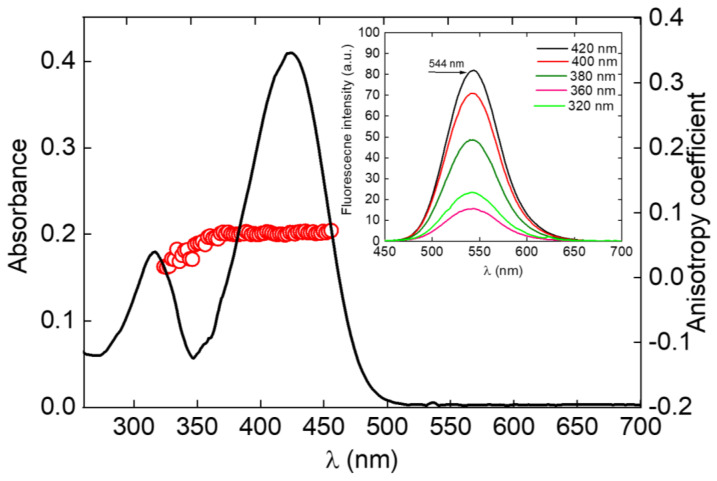
Fluorescence anisotropy (red circles) for compound **ChN(CH3)2**. The inset displays the fluorescence spectrum for different excitation wavelengths.

**Figure 5 molecules-28-01572-f005:**
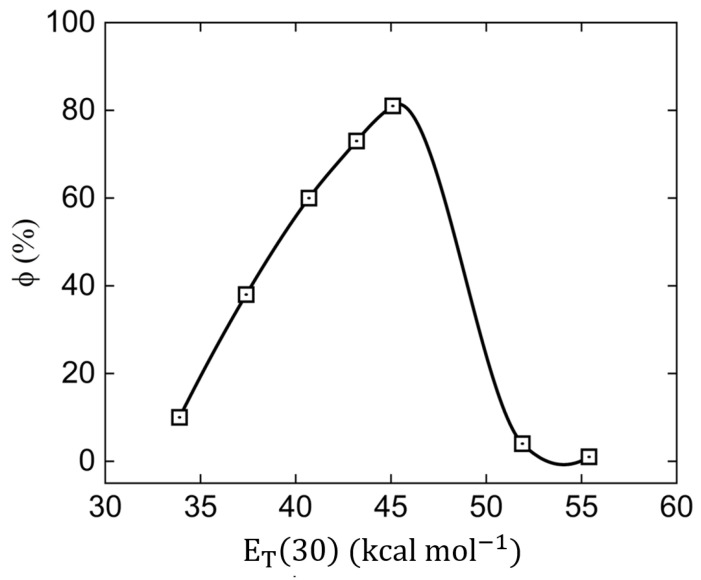
Fluorescence quantum yields of **ChN(CH3)2** for different solvents (in ascending order of ET(30): toluene, tetrahydrofuran, dichloromethane, dimethylsulfoxide, dimethylformamide, ethanol, and methanol) determined through the Brouwer method.

**Figure 6 molecules-28-01572-f006:**
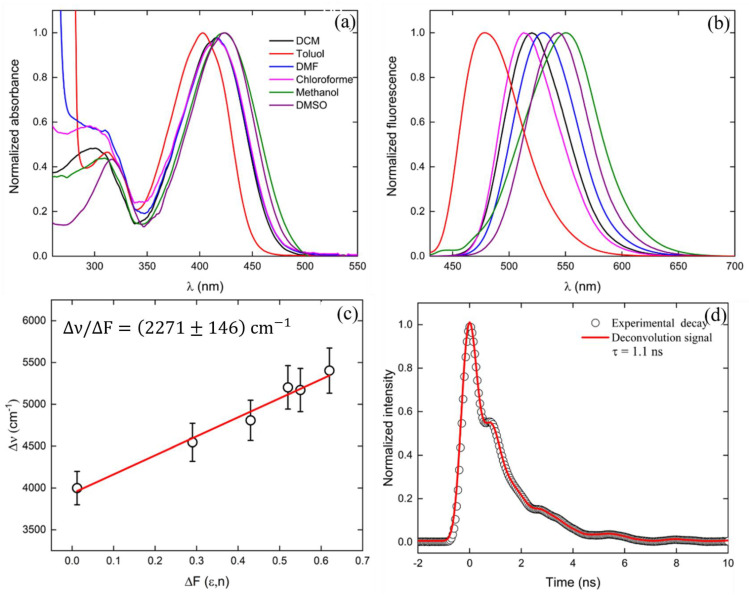
(**a**) linear absorption and (**b**) fluorescence emission of **ChN(CH3)2** for different surrounding media (solvents). (**c**) represents the Stoke Shift (in wavenumber units) for different values of solvent polarity (in ascending order of ΔF: toluene, chloroform, dichloromethane, dimethyl sulfoxide, dimethylformamide, and methanol), and (**d**) is the experimental (circles) fluorescence decay. The red line represents the adjustment considering the fluorescence experimental measurements and the instrument response function were convoluted.

**Figure 7 molecules-28-01572-f007:**
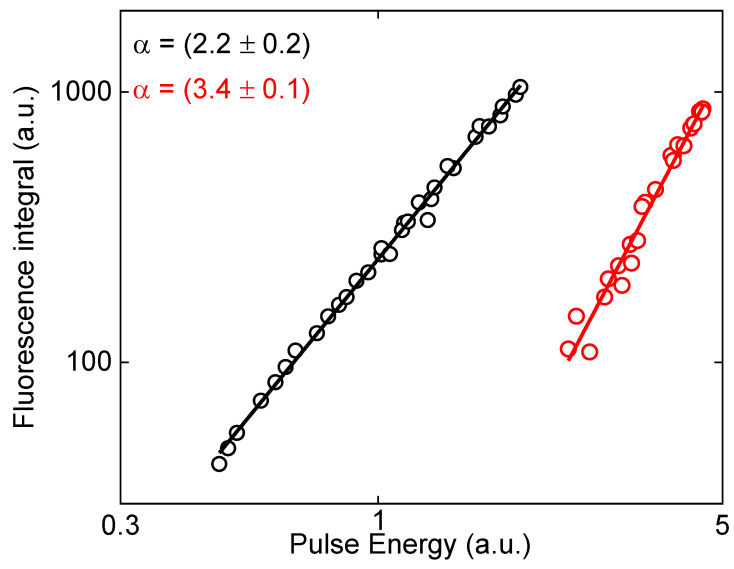
Quadratic (black circles) and cubic (red circles) dependence of the pulse energy with the fluorescence intensity confirming that the fluorescence of ChN(CH3)2 is triggered by two (900 nm) and three (1190 nm) photon excitations. The continuous lines represent the linear fitting.

**Table 1 molecules-28-01572-t001:** Values of $\varepsilon_{01}$, $\varepsilon_{02}$, 2PA cross-section at the lower ($\delta_{S1}^{2PA}$) and higher ($\delta_{S2}^{2PA}$) energy band for all compounds studied in this work. The wavelength of each electronic transition is also shown below. Values of 2PA cross-section are given in Goeppert-Mayer (1 GM = 10^−50^ cm^4^ s molecules^−1^ photons^−1^).

Compound	$\begin{array}{c} \varepsilon_{01}\left(M^{-1}cm^{-1}\right)\\ \lambda (nm) \end{array}$	$\begin{array}{c} \varepsilon_{02}\left(M^{-1}cm^{-1}\right)\\ \lambda (nm) \end{array}$	$\begin{array}{c} \delta_{S1}^{2PA}\left(GM\right)\\ \lambda (nm) \end{array}$	$\begin{array}{c} \delta_{S2}^{2PA}\left(GM\right)\\ \lambda (nm) \end{array}$
**ChH**	27,414 336	10,767 314	9 697	15 624
**ChCH3**	34,843 342	13,783 314	15 673	32 624
**ChOCH3**	30,661 355	14,512 319	20 695	33 634
**ChCH2CH3**	34,312 342	13,582 319	13 695	33 634
**ChOCH2CH3**	33,113 355	13,311 319	18 706	53 624
**ChN(CH3)2**	35,168 422	16,331 314	54 840	32 620
**ChBr**	34,164 336	14,378 309	8 697	33 624
**ChNO2**	21,665 342	13,852 294	15 706	36 555

**Table 2 molecules-28-01572-t002:** Photophysical parameters used as input parameters in the SOS approach and transition dipole moment from the first to the second excited state ($\mu_{12}$) determined as adjustable parameters in the SOS approach. $\Delta \mu_{01}^{G}$ and $\Delta \mu_{02}^{G}$ represent the value of $\Delta \mu_{01}$ obtained through QCCs. The asterisk indicates the value experimentally determined and previously discussed in this work.

Compound	μ_01_ (D)	μ_02_ (D)	$\Delta \mu_{01}^{G}$	$\Delta \mu_{02}^{G}$
**ChH**	5.9	3.3	2.5	6.1
**ChCH3**	6.5	4.2	4.6	7.7
**ChOCH3**	6.3	4.9	3.1	9.4
**ChCH2CH3**	6.4	3.9	4.8	8.0
**ChOCH2CH3**	6.7	4.5	3.1	9.3
**ChN(CH3)2**	7.5	4.7	11.5, 7.1 *	10.2
**ChBr**	6.5	4.3	3.1	7.0
**ChNO2**	6.3	5.9	8.2	8.2

**Table 3 molecules-28-01572-t003:** Acetamide-chalcone derivatives synthesized by Claisen-Schmidt condensation and the reaction yield, melting points, and visual colors of products. The benzene ring linked to the aceto-amide group was named ring A, and the one with peripherical groups was named ring B.

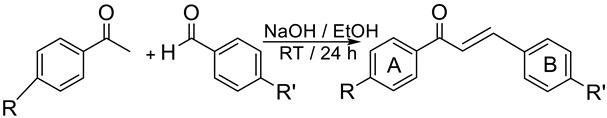
**R-**	**R’-**	**Yield**	**Melting Point**	**Color**
CH_3_COHN-	-H	30%	162.5–163.1 °C	Pale yellow
CH_3_COHN-	-CH_3_	72%	197.8–199.9 °C	Bright yellow
CH_3_COHN-	-CH_2_CH_3_	46%	200.3–201.6 °C	Bright yellow
CH_3_COHN-	-OCH_3_	41%	204.2–205.7 °C	Pale yellow
CH_3_COHN-	-OCH_2_CH_3_	42%	134.6–135.8 °C	Pale yellow
CH_3_COHN-	-N(CH_3_)_2_	30%	150.5–152.1 °C	Orangish
CH_3_COHN-	-Br	74%	221.7–222.1 °C	Yellow
CH_3_COHN-	-NO_2_	27%	234.0–237.0 °C	Bright yellow

## Data Availability

The data presented in this study are available on request from the corresponding author.

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
