# Peer review of "Two-Photon Absorption and Multiphoton Excited Fluorescence of Acetamide-Chalcone Derivatives: The Role of Dimethylamine Group on the Nonlinear Optical and Photophysical Properties"

_molecules, 2023, doi:10.3390/molecules28041572_

Round 1

Reviewer 1 Report

The introduction, written factually, clearly presents the idea of ​​conducting the described research as well as the subject matter. Results are presented in an accessible way, a properly conducted discussion. Materials and methods and SI part to be completed and improved. Conclusions clearly summarize the article.

Major remarks:

1) Figure 1: Please correct it because it is blurry.

2) Figure 2: Please mark the compounds, i.e. ChH, ChCH3, etc. on the graphs, eg in the upper left corner of each graph. This will make it easier to understand the figure without reading the description below it.

3) Please remove unnecessary blank lines between paragraphs, eg lines 150-151, etc. In addition, please standardize the line spacing throughout the article.

4) Why was DMSO chosen for the study and not another solvent?

5) How the extension of the alkyl chain, in analogs to the compound ChN(CH3)2, like ChN(CH2CH3)2, etc., would affect the properties of the compound(s).

6) Please add a brief general description of the synthesis of the compounds in Section 4.

7) In Table 3, please write the specific base instead of "OH-" (above the arrow in the reaction) and the conditions (heating, temperature?, RT) and reaction time). Apart from the substituents, the table does not present anything new. You might as well give an explanation of R and R' under the reaction scheme instead. A good addition to the table, if the authors want to leave it, would be yields.

8) Section 4: Please add manufacturers, cities and countries of the equipment used.

9) Why were the 13C NMR and HRESI-MS spectra of the compounds not performed? It is probably obvious that for compounds a full spectroscopic characterization should be performed. Please complete these data and put all spectra in SI.

10) Section 4 and SI: should be "1H NMR" instead of "1H NMR".

11) Line 372: should be "CDCl3" instead of CDCl3" and "DMSO-d6" instead of "DMSO-d6"

12) Lines 374-382: please correct it because you can see that as a result of formatting it got messed up

13) The authors in line 55 write: "a novel acetoamide-chalcones" which is quite interesting because the compounds have CAS numbers and are listed in the SciFinder database. Please clarify and remove the word "novel", as the presented compounds are not new. Additionally, there should be "acetamide" instead of "acetoamide".

14) In SI, please add in what form and color each compound was.

15) Please add melting points for each compound in SI.

16) SI: "RMN" - please expand the abbreviation. Should not there be NMR?

17) SI: last compound - please write the spectroscopic data in normal font, not in italics.

Author Response

Please find enclosed the revision of the manuscript “Two-photon absorption and multiphoton excited fluorescence of acetamide-chalcone derivatives: the role of dimethylamine group on the nonlinear optical and photophysical properties”. We have taken into consideration all the reviewer's comments and suggestions and modified the manuscript accordingly. All changes are highlighted in the revised version of the manuscript.

If you have any questions, feel free contacting me.

Prof. Cleber Renato Mendonca

Reviewer 2 Report

This manuscript reports using spectroscopic measurements and quantum chemistry calculations to study effects of electron-withdrawing and electron-donating groups on the linear and nonlinear optical properties of several acetamide-chalcone derivatives. The result indicated substitution with dimethylamine group may lead to fluorescence emission with high yield that can be induced by two and three photons excitations. It also found that this substitution can red-shift the lowest energy absorption band and increase strongly its two-photon absorption cross-section compared to the other derivatives. In addition, the result showed that all the compounds examined displays low energy excited states accessible by two photon absorption likely due to coupling of these states with the ground state.

This work is interesting in light of the various applications of the compounds examined and the fundamental importance of the substitution effects in altering photo-physical and nonlinear optical properties of the derivatives. The methods used are largely proper and the data obtained in reasonable quality. However, some of the analysis and interpretation of the data appears not sufficient. For improvement, the authors are suggested to consider the following comments:

1.      Page 8, Figure 4: why the data of fluorescence anisotropy was plot in the wavelength region for the absorption spectrum instead of the fluorescence spectrum? The values of the fluorescence anisotropy are between 0.1-1 and nearly independent of the wavelength at ~380-450 nm. What is the interpretation of this observation in terms of identifying nature of excited state responsible for the fluorescence emission? Method for measuring the fluorescence anisotropy should be added to section 4 (Materials and Methods).

2.      Page 9, Figure 5: the fluorescence yield is said to be lower in non-polar solvent and this was connected to the proximity effect of np* and pp* state. Is the np* state observed in the TDDFT calculation for absorption (like in Figure 3)? What is the origin of the lone pair electrons of the np* state? Is there other explanation to the different yields in the varied solvents? The solvents used for the fluorescence yield measurement should be specified in the figure caption.

3.      Page 10, Figure 6a: the absorption spectra in the different solvents should be displayed with the Y axis in extinction coefficient to better show influence of the solvents on the absorption transitions. Figure 6c: the solvents used for gaining the data points should be specified in the figure caption. If solute-solvent hydrogen bonding is involved and responsible for the very low fluorescence yield in MeOH and EtOH, the Stokes Shifts in these solvents are expected to also deviate from those in the other solvents. The authors are suggested to examine and confirm if this was the case and to explain their data accordingly. Figure 6d: the decay profile appears to be complex, containing several components. How and what is the fluorescence lifetime derived from this decay profile? What is the instrumental response function of the measurement? The methods for measuring the fluorescence decay should be provided in section 4 (Materials and Methods).

4.      Is there involvement of photochemical reaction for ChN(CH3)2 in DMSO or the other solvents examined?

5.      Page 13, lines 352-355: the authors proposed ChN(CH3)2 could be “an outstanding candidate to be employed in biological applications”. For such a claim to hold, the authors are suggested to examine the linear and non-linear optical property of ChN(CH3)2 in water, an important and physiologically relevant solvent.                     

Typos:

1.      Page 3, line 88: “ε spectrarevels an intense absorption”.

2.      Page 3, line 92: “ChOCH3 and ChOCH2CH3 present the most red-shift effects” is confusing for the lowest energy absorption of ChN(CH3)2 is centered at c.a. 420 nm, more red-shifted than the absorption from ChOCH3 and ChOCH2CH3 at ~335 nm (Table 1).

3.      Page 5, line 144 “which was c.a. 840 nm),” and line 145 “which was c.a. 555 nm).”. Where is the other half of the bracket in the two sentences?

              4. Page 15, line 411-412 “For this purpose, was used the CAM-B3LYP functional……”

Author Response

Please find enclosed the revision of the manuscript “Two-photon absorption and multiphoton excited fluorescence of acetamide-chalcone derivatives: the role of dimethylamine group on the nonlinear optical and photophysical properties”. We have taken into consideration all the reviewer's comments and suggestions and modified the manuscript accordingly. All changes are highlighted in the revised version of the manuscript.

If you have any questions, feel free contacting me.

Prof. Cleber Renato Mendonça

Reviewer 3 Report

The authors have submitted a manuscript entitled " Two-photon absorption and multiphoton excited fluorescence of acetamide-chalcone derivatives: the role of dimethylamine group on the nonlinear optical and photophysical properties " to the journal. The authors present the synthesis and characterization of chalcone derivatives with nonlinear optical properties. In this work, some photochemical techniques are used to reveal the two- or three-photon absorptions of chalcone derivatives. It seems quite interesting that the two-photon absorption properties of acetamide-chalcone derivatives and the experimental results seem to have good quality. However, this work seems more suitable for other professional journals because it only deals with deep photochemistry and seems far from the general readership of the Molecules. The studies containing only ‘photochemistry’ are of little interest to the general reader except when the novel derivatives have very unique properties.

I have few comments:

1.     Basically, i have doubts about whether your chalcone derivatives are properly synthesized with high purity because there are only proton NMR and FT-IR results. The authors should add both HRMS and elemental analysis results of synthesized chalcone derivatives at least.

2.     What is the novelty of this work? chalcone-derivatives with having similar structures are already kwnon (J. Phys. Chem. A 2015, 119, 11128−11137) and chalcone-derivatives with nonlinear properties (two-photon absorption) are already reported (J. Phys. Chem. A 2020, 124, 51, 10808–10816). Moreover, introduction of electron withdrawing or donating groups group into backbone structure has been reported as a common molecular design strategy for decades.

3.     Considering numerous backbone structures such as diketopyrrolopyrrole, BODIPY, carborane, perylene derivatives with nonlinear optical properties, what is the advantages of chalcone derivatives than the other structures?

4.     Page 3, lines 92-92, the author described that ‘It should be highlighted that ChOCH3 and ChOCH2CH3 present the most red-shift effects, indicating that these groups may be affecting their molecular planarity’. What is the possible change of planarity or stacking? I also read the reference you provided (The Influence of Methoxy and Ethoxy Groups on Supramolecular Arrangement of Two Methoxy-chalcones, J. Braz. Chem. Soc. (2017). https://doi.org/10.21577/0103-5053.20170067), however, i could not find any clues that the planarity is related to the spectral red shit of ChOCH3 and ChOCH2CH3 than the others. In my opinion, the spectral red shifts of methoxy- or ethoxy-substituted molecules usually originated from the strong electron donating power by non-bonding electrons that exist around methoxy- or ethoxy-groups.

5.     In addition, what is the potential advantages of the increase of 2PA intensities? What is the highest 2PA intensities of molecules known so far? The authors can explain the relationship between 2PA intensity and some enhancement of properties in biological applications?

6.     In page 5, lines 152-166, the authors insisted that the synthesized chalcone derivatives in this work have greater 2PA intensities (GM unit) than similar structures which was already known in other literatures. However, those indirect comparisons with numerical values seem not suitable in this case because the experimental setup in this work are not completely identical to those literatures. Please conduct direct comparison using those known structures by synthesizing them.

7.     In my opinion, the acetamide-group on the A ring induces more interesting properties such as 2PA intensity increases than dimethylamine group on B ring. However, the authors not focused on the acetamide-group to explain the optical properties of chalcone derivatives and only focused on the substituents on B ring. This lack should be addressed by conducting additional experiments or descriptions.

8.     Among the synthesized chalcone structure, what is the possible reasons that only ChN(CH3)2 shows fluorescence or what is the possible reasons that the other chalcone derivatives show fluorescence quenching? The optical properties of chalcone derivatives in DMSO are only displayed in this work. The authors analyzed the optical properties of the chalcone derivatives in other solvent? The fluorescence is easily affected by solvent polarity and solvent relaxations. Those solvent effects highly affect the intramolecular charge transfer of the electron donor-acceptor structure. Therefore, the author should display the optical properties of other chalcone derivatives in other solvents with different polarity.

9.     The authors calculated the μ12 by mathematically but there are no direct results. The authors can experimentally demonstrate the transition of first excited state to second excited state? This can be observed in excited state absorption spectra (ESA) using transient state spectroscopy because the optical absorption is very fast transition (< 10-15 sec).

10.   In page 13 lines 354-355, the authors described that ‘it is an outstanding candidate to be employed in biological applications’. However, there are no simple clues that those are really outstanding candidate for biological applications such as cell toxicity tests or bio-imaging results.

11.   In page 15, lines 405-407, the author described ‘Vibrational frequency calculations were performed along with this calculation to confirm that the global minimum structures were reached’. How the authors can convince the global minimum not local minimum only from the calculated frequency results?

Author Response

(The authors gave the same response as above.)

Round 2

Reviewer 1 Report

Thank you very much for all the answears.

The changes introduced significantly increased the value of the manuscript.

A few final remarks:

1) Figure 5 - it is not clear which point refers to which solvent.

2) Lines 371-372 and 374-375 - solvent names should be written in lower case and in the case of solvent mixtures please add "(v/v)"

Author Response

Please find enclosed the revision of the manuscript “Two-photon absorption and multiphoton excited fluorescence of acetamide-chalcone derivatives: the role of dimethylamine group on the nonlinear optical and photophysical properties”. We have taken into consideration all the reviewer's comments and suggestions and modified the manuscript accordingly. All changes are highlighted in the revised version of the manuscript.

Prof. Cleber Renato Mendonça

Reviewer 3 Report

I read all the response letters including for the other reviewers. The reviewers raised some questions to the authors, and in some point of view, the authors may tried to address the issues. See the comments as below:

If there are any changes in the revised manuscript by comments of reviewers, please provide a point by point response to those of any reviewers: what changes were made for what reasons in where on which pages and lines in revised manuscript. Reviewing papers is voluntary work. The reviewers are not idle or kind enough to compare each revised points in between original and revised manuscript.

In question#6 and question #9, the author responded as ‘Two-photon absorption cross-section can be compared, even if different experimental methods are used, because it is an intrinsic molecular parameter.’ and ‘In fact, through transient absorption measurements it is possible to experimentally determine the value of μ12. However, to determine this parameter we need to probe the first excited state with a white light in the infra-red region (to match the energy difference between these states). In our case, we count with a shappire crystal as to produce white-light continuum, giving a range from 400 nm up to 750 nm, that just probes more energetic excited states.’, respectively. However, i cannot understand this response. If you cannot perform suitable revisions or additional experiments from the request of reviewers, please provide more reasonable and detailed explanations with appropriate citations in response letter rather than just providing short sentences meaning 'this is not necessary' to the reviewers.

In addition, appropriate citations ‘must’ be added if there is not enough experimental evidence for your claim. ex: ‘In addition, the red shift observed for these compounds could be associated to the 97 non-bonding electrons of methorxy and ethoxy groups.’ as newly added in page 3 lines 97-98.

Author Response

(The authors gave the same response as above.)
